# Semantic influences on object detection: Drift diffusion modeling provides insights regarding mechanism

Jingming Xue[1]*, Robert C. Wilson[1], Mary A. Peterson[2,3]

1 School of Psychology, Georgia Institute of Technology, Atlanta, Georgia, United States of America,
2 Department of Psychology, University of Arizona, Tucson, Arizona, United States of America,
3 Cognitive Science Program, University of Arizona, Tucson, Arizona, United States of America

* jxue93@gatech.edu

## Abstract

Research shows that semantics, activated by words, impacts object detection. Skocypec & Peterson (2022) indexed object detection via correct reports of where figures lie in bipartite displays depicting familiar objects on one side of a border. They reported 2 studies with intermixed Valid and Invalid labels shown before test displays and a third, control, study. Valid labels denoted display objects. Invalid labels denoted unrelated objects in a different or the same superordinate-level category in studies 1 & 2, respectively. We used drift diffusion modeling (DDM) to elucidate the mechanisms of their results. DDM revealed that, following Valid labels, drift rate toward the correct decision increased, i.e., SNR increased. Invalid labels do not affect drift rate directly, but they create a context that diminishes the facilitative effect of valid labels on evidence accumulation. Threshold was higher in study 2 than control, but not in study 1. That more evidence must be accumulated from displays that follow labels denoting objects in the same-superordinate category as the object in the display indicates that more evidence from the display is needed to resolve semantic uncertainty regarding which object is present. These results support the view that semantic networks are engaged in object detection.

## Author summary

Object detection entails the first awareness of the presence of a shaped entity. We found that detecting where a known object lies relative to a border was enhanced when valid word labels preceded test displays. In contrast, accurate detection was delayed by invalid labels denoting an object in the same superordinate-level category as the target but not by invalid labels denoting an object in a different superordinate-level category. We used a drift diffusion model to reveal the mechanisms of these results. The model showed that (1) the

**Data availability statement:** Data and code to replicate these analyses are available at https://osf.io/g3mxb/.

**Funding:** The author(s) received no specific funding for this work.

**Competing interests:** The authors have declared that no competing interests exist.

signal-to-noise ratio (drift rate toward an accurate decision) was higher following valid labels and (2) the amount of evidence that had to be accumulated from the display before accurate detection was higher following same superordinate-level category invalid labels. These results suggest that labels affect more than feature expectations; they alter activation in a semantic network involved in object detection.

## Introduction

A long-standing question regarding visual perception is whether knowledge such as the meaning of words (i.e., semantics) can influence how we perceive objects [1–10]. For example, does receiving information about a stimulus beforehand make it easier to *recognize* it later? Previous work suggests that, for object **recognition**, the answer is yes [11–15]. Recent research has suggested that object **detection** – considered a more basic perceptual process than recognition – is also influenced by semantics [16–18]. These detection findings might suggest that high-level semantics influence perception far down the perceptual hierarchy. However, controversy remains over whether those previous studies assessed object detection *per se* or the detection of features of the denoted objects [19,20]. Moreover, if semantics does influence object detection, questions must be answered regarding how it does so. Whereas experimental results can support some hypotheses, modeling can further elucidate mechanism.

In the present article, we use computational modeling to reveal the mechanisms whereby semantics influenced object detection in a recent experiment by Skocypec and Peterson [19]. We begin by discussing the "Label-Feedback" hypothesis, a leading hypothesis regarding how semantics might influence object detection that was supported by previous experiments in which stimuli were repeated multiple times. In the next section, we discuss an alternative theoretical interpretation supported by Skocypec and Peterson's results [19]. We discuss Skocypec and Peterson's manipulations and results [19] in some detail before introducing drift diffusion modeling, the modeling method used in this article.

### A leading hypothesis: the label-feedback hypothesis

The "Label-Feedback" hypothesis regarding how semantics might influence object detection suggests that presenting a valid label for an object shortly before the object appears (e.g., the word "umbrella" before an image of an umbrella) activates representations of that object all the way down the visual hierarchy to low-level features. This pre-activation improves object detection by enabling faster progression through the visual hierarchy from lower-level features to higher-level object representations [21]. On this hypothesis, an invalid label would interfere with object detection by activating low-level features of a different object; mismatching input would necessitate revising those predictions which would slow progression through the visual hierarchy.

In studies that provided support for the Label-Feedback hypothesis, a basic-level label (i.e., a label commonly paired with an object) was paired with a particular exemplar many times [1,3,18]. However, with so much repetition, participants may have learned to respond accurately following a basic-level label when they detected low-level features rather than the object configuration [19,20]. Indeed, Lupyan and Ward [18] showed that word labels operate on feature representations rather than semantic representations. Thus, the use of repetition may produce a misunderstanding of how basic-level labels typically affect object detection.

### An alternative semantic network hypothesis

Skocypec and Peterson [19] proposed an alternative account of how basic-level word labels affect object detection: Without repetition, valid basic-level labels activate a distributed semantic network that represents words, objects, object properties, object affordances, commonly performed actions, etc. [22] Object representations in this network take the form of neural populations that are tuned by previous encounters with the objects ([23–26]; hence, more units are tuned to the typical, upright, configuration of the represented object than to atypical orientations. Therefore, activity in a neural population accumulates more quickly when an upright rather than an upside down, "inverted", object is encountered. If decisions about the presence of an object depend on accumulated evidence in the neural population selective for that object configuration per se rather than its features, then detection decisions would be faster and/or more accurate for upright than inverted objects [27,28]. In contrast, it is commonly accepted that detection decisions based on features should not vary with orientation because features alone do not activate configurations [19,20,29–31]. Therefore, one way to determine what type of representations mediate semantic effects on object detection is to investigate whether effects of labels depend on the orientation of the object denoted by the word. If low-level feature representations are responsible, the effects of word labels should be orientation independent. In contrast, if via a semantic network, the words activate neural populations representing the objects they denote, then the effects of word labels should be larger for upright than inverted objects.

### Skocypec and Peterson's (2022) study

Skocypec and Peterson [19] assessed object detection via figure assignment reports regarding stimuli like those in Fig 1. These sample stimuli were adapted from [32]. In these stimuli, a portion of a well-known (i.e., "familiar") object with a typical upright is depicted on one side of a central border that divides the display into two equal-area halves. The term "familiar" in this article refers to previous experience outside the laboratory with objects in the same basic-level category, not to repetition within the experiment. The objects depicted in the bipartite stimuli were new instances of well-known objects. They were not repeated in the experiments. In previous research Peterson and colleagues had found that figures (i.e., objects) are more likely to be detected on the side of the border where a familiar configuration is sketched when the display is upright than when it is inverted. For example, participants are more likely to perceive the object on the left in Fig 1A where a portion of a woman is depicted in a familiar upright configuration rather than in Fig 1B when the woman is depicted in an unfamiliar inverted orientation; the features are the same, but the configuration is different. Likewise, the figure/object is more likely to be detected on the right in Fig 1C than Fig 1D [26]. This orientation dependency implicates input to object detection decisions from neural populations representing the objects in these displays.

In their experiments, Skocypec and Peterson [19] investigated the effect of word labels on object detection in these bipartite displays. Participants in experimental labels-present groups were presented with either a valid or an invalid label before viewing a brief, masked, exposure of a bipartite display depicting a portion of a familiar object in either an upright or an inverted orientation. Valid labels denoted the object at a basic level (e.g., valid labels for the objects in Fig 1 were "woman" and "bell"). Skocypec and Peterson [19] leveraged the orientation dependence of familiar configuration effects to examine whether effects of valid word labels shown before bipartite displays are mediated by activation of low-level

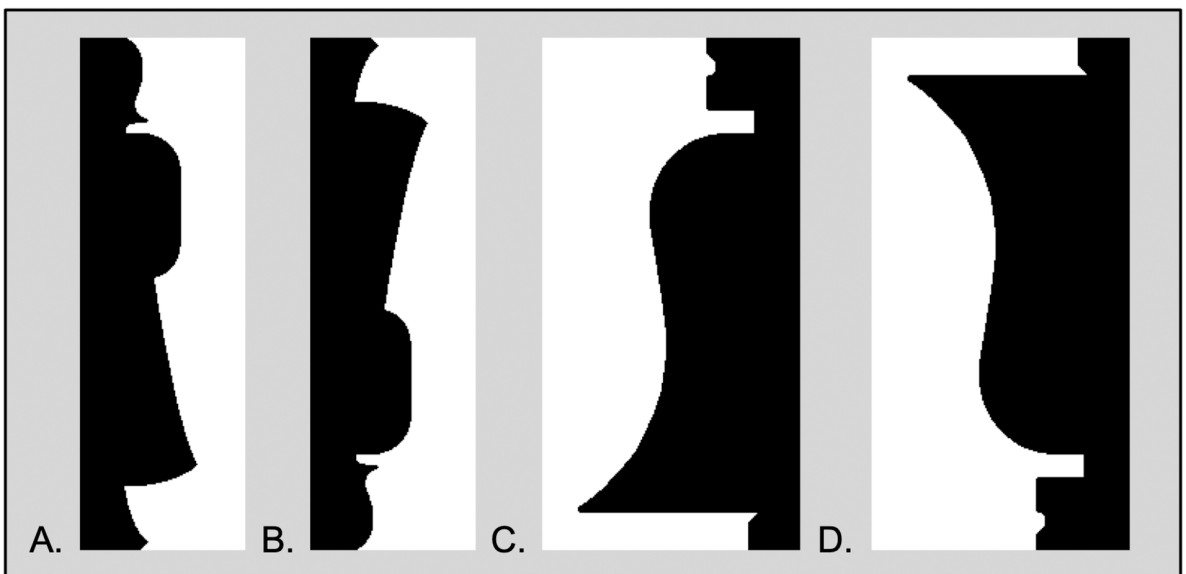

**Fig 1. Examples of bipartite stimuli.** A portion of a well-known object was sketched on one side of the central border of stimuli in which two equal-area regions, one black and one white lay on opposite sides of that border. One, "critical", region depicted a well-known object in an upright or an inverted orientation and was equally often on the left and right, in black and white. A. The critical region depicting a portion of a woman in an upright orientation is on the left in black. B. An inverted version of A. C. The critical region depicts a portion of a bell in an upright orientation in black on the right side. D. An inverted version of C. Black/white color of the critical regions was balanced in the experiments.

features or by higher-level neural populations representing configured objects. If low-level features drive the effect, label effects should be orientation invariant; if higher-level neural populations are key, label effects should be larger for upright than inverted objects. To be noted, individual participants viewed each bipartite display (upright or inverted) and each label once only to assess semantic influences unaffected by repetition.

Valid labels preceded the bipartite displays on half the trials; invalid labels denoting unrelated objects preceded the displays on the other half of the trials (label type was randomly intermixed). To ensure the words are unrelated to the objects in the target display, we used the database SUBTL Word Frequency Database [33]. The invalid labels used in study 1 and study 2 were different. In study 1, the invalid labels denoted an object from a **different** superordinate-level category, where the superordinate-level categories were natural versus artificial objects. For instance, the invalid labels for woman, a natural object, and bell, an artificial object, were, respectively, "money" (artificial) and "fish" (natural). In study 2, the invalid labels denoted an object from the **same** superordinate-level category as the object sketched in the upcoming display, albeit unrelated (e.g., "shark" (natural) and "book" (artificial) for woman and bell, respectively). To determine whether valid labels improved object detection, invalid labels impaired object detection, or whether both effects occurred, Skocypec and Peterson [19] also compared performance in these labels-present studies to performance in a control study in which the same displays were not preceded by words (labels-absent control).

Overall, Skocypec and Peterson's results [19] supported the hypothesis that effects of basic-level word labels shown once only are best explained by activation in semantic networks rather than by predictions regarding the low-level features of objects: In both studies, valid labels improved detection accuracy and response times (RTs) over control and effects were larger when the familiar objects in the bipartite stimuli were depicted in an upright rather than an inverted orientation (Fig 2). Furthermore, in study 1, different superordinate-level category invalid labels did not affect detection accuracy or response times compared to results obtained in the control labels-absent study. These results were also contrary to predictions from a Label-Feedback account where interference is expected from invalid predictions. In study 2, following

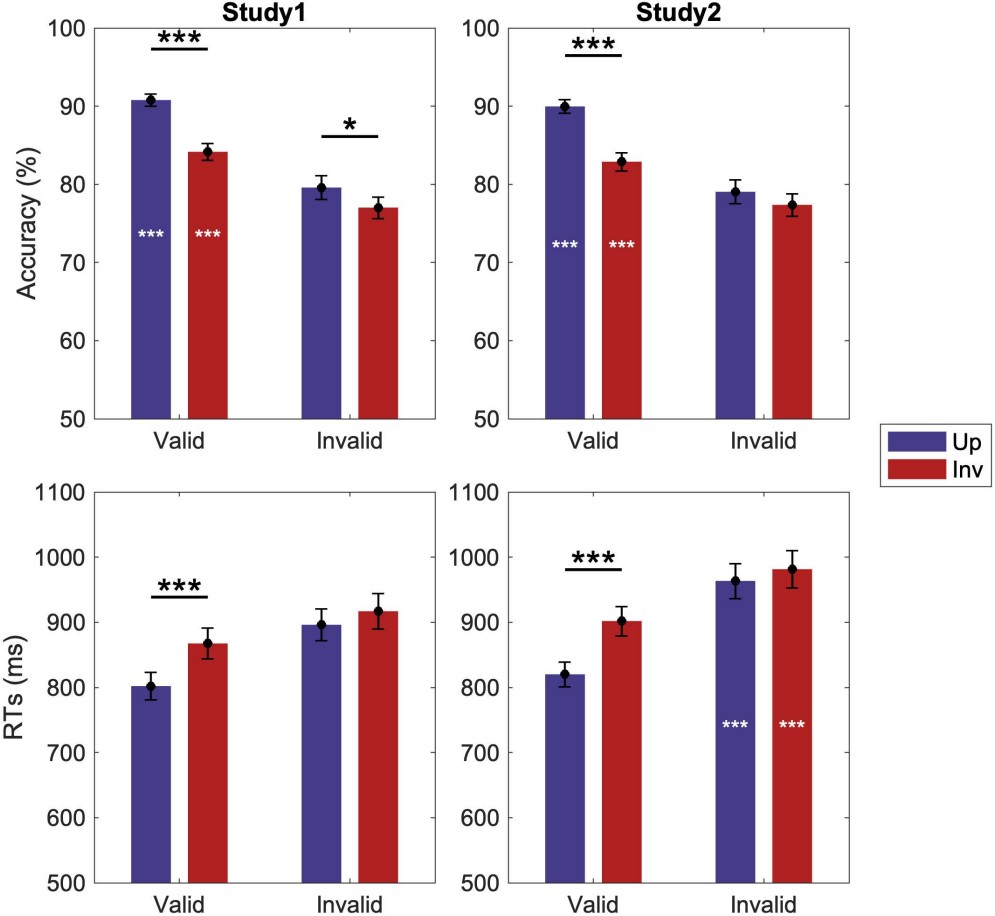

**Fig 2. Skocypec and Peterson's behavioral results [19].** (First column) study 1; (Second column) study 2. (First row) Accuracy; (Second row) response times. Blue bars: Performance with upright displays; Red bars: Performance with inverted displays. White asterisks indicate main effects of labels-present vs. labels-absent control groups. Horizontal lines and black asterisks indicate orientation-dependent differences. Error bars represent pooled standard errors. *** indicates $p < 0.001$, * indicates $p < 0.05$.

same superordinate-level category invalid labels, however, detection RTs were substantially and significantly increased compared to control for both upright and inverted objects. This result was unexpected on the Label-Feedback hypothesis because fewer prediction revisions should be necessary when invalid labels denote objects in the same rather than a different superordinate-level category. Skocypec and Peterson [19] attributed these results to a conflict in the overlapping semantic networks activated by the invalid label and the object in the test display, given that objects in the same superordinate-level category are highly likely to share properties, features, potential actions, etc. [34]. Skocypec and Peterson [19] hypothesized that this conflict had to be resolved before detection occurred.

Skocypec and Peterson [19] interpreted their result as evidence that object detection is not simply affected by semantic activation; it *entails* semantic activation. This was a new proposal. Here we use computational modeling to elucidate the mechanisms of Skocypec and Peterson's results [19].

## The present research

We used a drift diffusion model [35,36] to reveal the underlying cognitive computations involved in detecting the objects in the bipartite displays in Skocypec and Peterson's experiments [19]. By jointly modeling object detection accuracy and RT

on participants' individual trials (rather than mean RTs per participant per condition like Skocypec and Peterson [19]), this model can tell us whether the object detection decision is controlled by a change in *drift rate* (roughly processing speed), a change in evidence *threshold*, a change in starting point (here, a left/right side bias), or a change in non-decision time. Drift diffusion models have proven valuable for studying neural mechanisms across a wide range of species and contexts [37–43], with their parameters known to associate with neuronal activity [44]. Here, we investigate whether applying a Drift Diffusion Model (DDM) to figure-ground perception—for the first time—can capture Skocypec and Peterson's results [19]. By doing this, we aim to provide a quantitative description of the orientation effect through drift rate analysis. Additionally, this novel application of modeling allows us, for the first time, to extract insights into key parameters such as non-decision time and boundary separation (i.e., threshold), deepening our understanding of the underlying cognitive processes. Lastly, we can also test whether there is any additional process that the model cannot identify, such as lapses of attention.

Our drift diffusion model (DDM) assumes that, when presented with a bipartite test display, participants make a left/right response by accumulating "evidence" over time in favor of one response or the other. This evidence is assumed to contain both signal – whether the object lies to the left or right of the central border – as well as noise caused, e.g., by ambiguous cues and random neuronal responding. Thus, the accumulated evidence, $y$, evolves according to a drifting random walk with a noise parameter, $s$, controlling the randomness of the walk and a drift parameter, $v$, controlling the drift (see Fig 3A). In line with common practice, we set $s = 1$.

In Fig 3B, a positive drift rate denotes evidence that the object is on the right and the evidence tends to drift upwards over time. Conversely, a negative drift rate corresponds to the case in which the object is on the left; here, evidence tends to drift down (Fig 3C). The relative magnitude of the drift rate to the noise (i.e., $|v|/s$) denotes the "signal-to-noise ratio," with a higher ratio implying stronger evidence and faster processing and a lower ratio implying weaker evidence and slower processing (Fig 3D). An initial condition, $z$, controls the initial bias for one option or the other (e.g., a bias for responding right/left, Fig 3F and 3G).

The model makes a decision when the evidence crosses one of two "decision thresholds" at $y = 0$ or $y = a$. In Fig 3B and 3C, the model chooses "right side" when it crosses the positive threshold and "left side" when it crosses the negative threshold. The time at which the model crosses the threshold is known as the Decision Time (DT), the time taken by the decision process. This Decision Time is further related to the Response Time (RT) through the addition of a non-decision time $T_{er}$, which captures sensory, motor, and other non-decision related delays between the onset of the sensory stimulus and the response. Thus, by modeling which threshold is crossed the DDM can model the accuracy and by modeling when the thresholds are crossed the DDM can model the response times. By fitting the model to human choices and response times, we can therefore estimate the four free parameters of the model – drift rate, $v$; the initial condition, $z$; the threshold, $a$; and non-decision time, $T_{er}$ – and investigate how they change in different label conditions. The noise parameter in the model is fixed at a value of 1. This parameter is integrated with the drift rate to determine the signal-to-noise ratio. For further details, please refer to the methods section.

### Mapping the drift diffusion model to Skocypec and Peterson's (2022) results

By identifying the parameters of the label effects in a DDM, we aim to test Skocypec and Peterson's [19] Semantic Network interpretation of their results. First, we can test their hypothesis that the orientation effects following valid labels occur because the valid label operates through a semantic network to pre-activate the neural population representing the familiar object it denotes. On this hypothesis, the drift diffusion model should return a higher drift rate for the upright than the inverted condition when valid word labels precede the displays compared to the control condition. Second, we can assess the mechanisms of the different invalid label effects observed in study1 and study 2. Are the differences due to changes in drift rate or changes in threshold or both? We will also investigate whether there are study-dependent starting point differences to investigate whether participants have an a priori bias to press the right or left button and non-response time, $T_{er}$, to examine whether stimulus encoding or response execution processes are affected by the labels shown before the test displays.

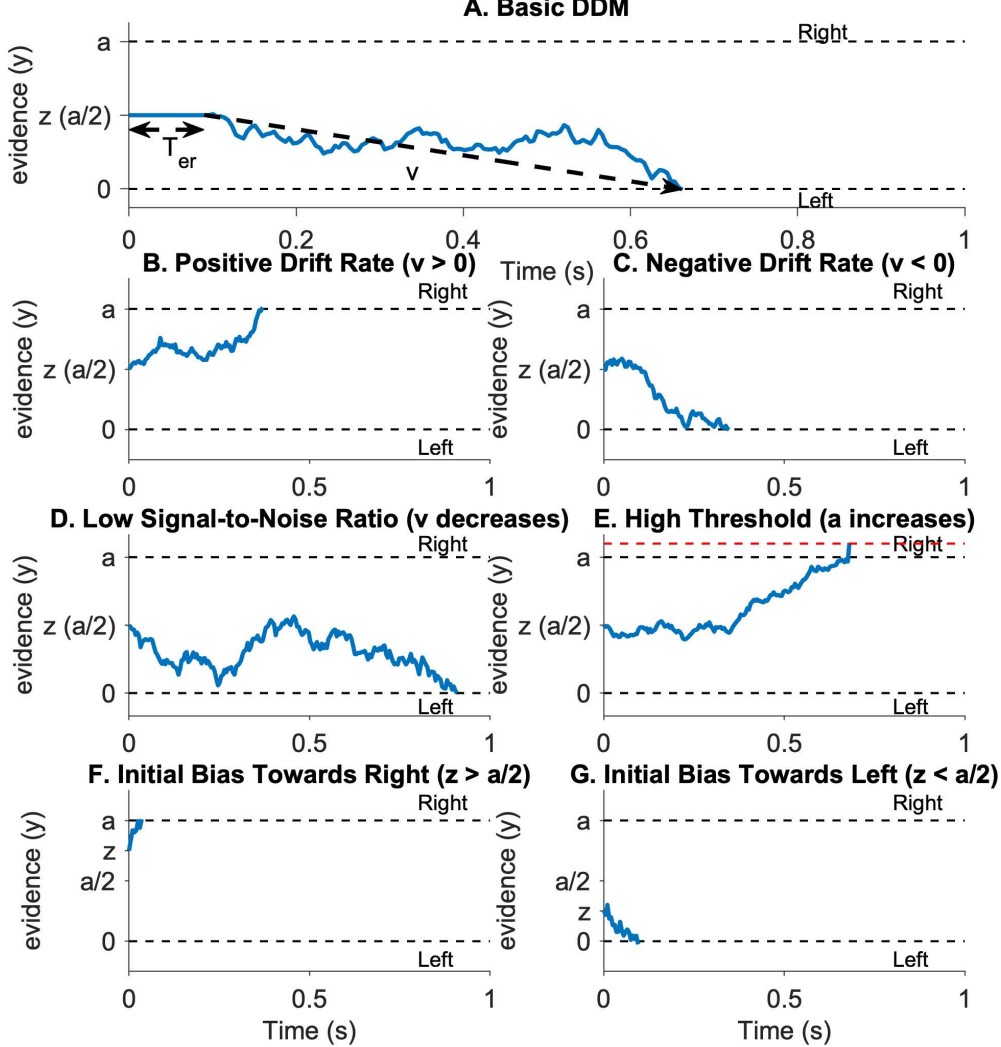

**Fig 3. Different cases of drift diffusion model.** (A) A basic drift diffusion model over time. The x-axis represents time in seconds, and the y-axis represents evidence accumulation, which could be a cognitive or a perceptual process leading towards a decision. The dashed lines at the top and bottom represent decision thresholds for two choices, labeled as "Right" and "Left", denoting by $0$ and $a$ separately. Evidence accumulates over time, starting from $z$ and fluctuating until it reaches one of the thresholds, indicating a decision has been made. The double headed arrow denotes the non-decision time, $T_{er}$. In A, the evidence crosses the upper threshold, suggesting a decision made towards the "Right side" option. (B) A decision process with a positive drift rate, $v$, on evidence accumulation ($v > 0$). A positive drift rate indicates a tendency for the evidence to accumulate towards the "Right side" choice over time. (C) A negative drift rate ($v < 0$) indicates a tendency to accumulate evidence towards the "Left side" choice. (D) A lower signal-to-noise ratio where there is more variability and noise in the evidence accumulation process than in C, resulting in a longer path to a decision threshold. (E) A higher threshold condition where more evidence must be accumulated before decision. (F) Evidence accumulation begins with a bias towards the "Right" choice, starting above zero, indicating a predisposition towards that option. (G) The opposite initial bias towards the "Left" choice, with evidence accumulation starting below zero.

## Method

### Ethics statement

The Institutional Review Board of the Human Subjects Protection Program at the University of Arizona approved the procedures in this study, including the consent form. Project title: "Multiple Spatial and Temporal Scales: Experimental

Investigation And Computational Modeling". Project Number: 1404304055. Written formal consent was obtained from all participants.

We applied our DDM to the response times for both accurate and inaccurate responses recorded by Skocypec and Peterson [19]. In what follows, we briefly describe their method in more detail.

## Participants

Data were analyzed from 321 undergraduate students (18–36 years old; M = 19.21, SD = 1.85; 236 F and 85 M) from the University of Arizona (UA) who participated for course credit or payment. The total included participants who participated in one of 12 experiments: an original and a replication for each of two exposure durations (90 ms and 100 ms) in either the control no-labels study; study 1 (a labels-present study with different superordinate-level category invalid labels intermixed with valid labels); or study 2 (a labels-present study with same superordinate-level category invalid labels intermixed with valid labels)). Skocypec and Peterson [19] found no substantive differences between the two exposure conditions; hence the results are combined here. All participants provided informed consent and had normal or corrected-to-normal vision. (For more information about participants see Peterson and Skocypec [19].

## Test displays/stimuli

The stimuli were bipartite displays, in which a central border divided a vertically elongated rectangle into two equal-area regions (samples shown in Fig 1), one black and one white. The central border sketched a portion of a well-known object (*a familiar configuration*) on one side; this was the "critical side". The critical side was equally often on the left/right and colored black/white. The stimuli were shown on a medium gray backdrop. All three studies used the same 72 displays, including 36 upright and 36 inverted displays. In upright displays, the critical side sketched a familiar object its typical upright orientation. The inverted displays were obtained by rotating the upright display 180° and mirroring them on the vertical axis. Half of the familiar configurations depicted portions of natural objects, and the other half depicted portions of artificial objects. The side complementary to the critical side depicted a novel shape. (For normative data regarding these stimuli, see [32] Every participant saw each display once only. Stimuli were balanced across conditions in 16 programs, each seen by eight participants per study. More information about the displays can be found in Peterson and Skocypec [19]; the displays can be accessed at https://osf.io/j9kz2/.

## Procedure

First, participants were introduced to the object detection task; they were told that their task was to report whether they detected a shaped entity -- a figure – on the left or right side of the central border by pressing a left (right) button on a response box. They made their response with the index and middle fingers of their dominant hand. They were instructed to report their first impression of where they detected a figure. This was our assay of object detection. We scored their responses as "correct" if they reported a figure on the familiar configuration side of the border and as "incorrect" if they reported a figure on the other side of the border. Participants were unaware of this scoring procedure; they were also told that there were no correct or incorrect responses; that we were interested in what they saw. Responses made up to 4000 ms after test display onset were recorded.

The test procedure is shown in Fig 4. Each trial started with a central fixation cross. Participants were instructed to press a foot pedal when they were ready for the trial to begin. In studies 1 and 2 (the labels-present studies), after the foot pedal was pressed, a label was shown for 250 ms followed by a blank screen for 500 ms. Next, a bipartite display was shown briefly; different groups viewed the displays for 90 or 100 ms. (Skocypec and Peterson [19] found that exposure duration did not differentially affect accuracy in the labels-present studies versus the control study. Hence, the present analysis uses data from the two exposure durations.) The bipartite display was followed immediately by a mask (200 ms) and then by a blank screen that was present until response or 4000 ms had elapsed. Participants' RT was recorded from

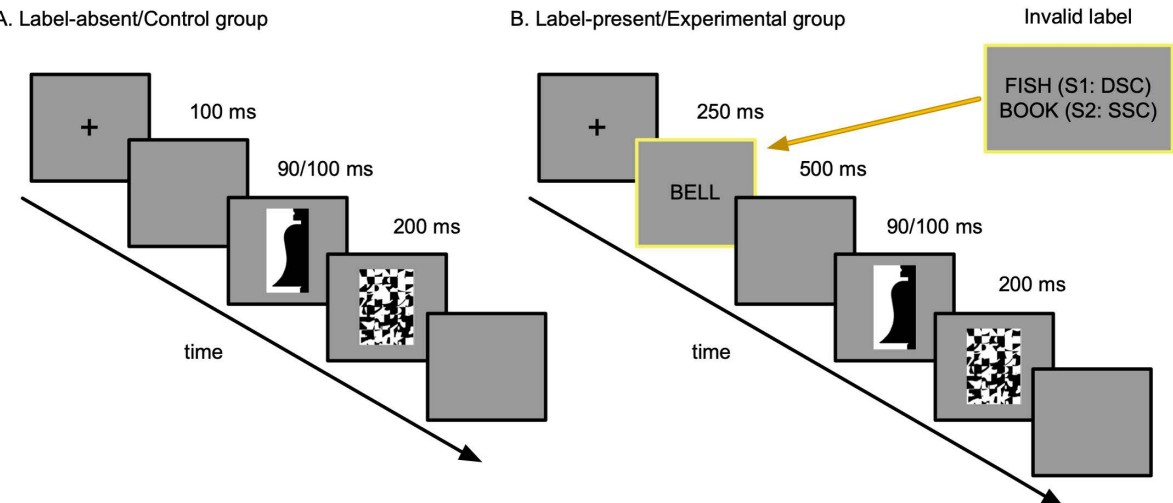

**Fig 4. Trial structures for labels-present and labels-absent studies.** (A) Trial structure for labels absent experiments (control studies). Each trial began with a fixation point shown on a medium gray background (which was used throughout), followed by a 100 ms blank screen. Then, the stimulus was presented for 90 ms or 100 ms to different groups, followed by a mask for 200 ms and then by a blank screen. (B) Trial structures for labels present experiments, the critical difference from control is that, after the fixation cross, a word label appeared in the center of the screen for 250 ms. The label was either valid, denoting the object in the upcoming bipartite display at a basic level, or invalid, denoting a different unrelated object. Invalid labels in study 1 denoted an object in a different superordinate-level category (i.e., natural vs. artificial), denoted by "DSC" in the figure; invalid labels in study 2 denoted an object in the same superordinate-level category, denoted by "SSC" in the figure. The inset shows sample invalid labels for the object depicted in the bipartite display (i.e., bell) in study 1 (S1) and study 2 (S2). For a complete listing of the invalid labels see the Appendix in Skocypec and Peterson [19].

the onset of the bipartite display. Half of the labels in studies 1 and 2 were valid in that they were basic-level category labels for the object sketched on the critical side of the central border in the upcoming display. The other half were invalid in that they denoted a different object. Both studies used the same valid labels. The invalid labels were different in the two studies: In study 1 the invalid labels denoted an unrelated object from a **different** superordinate-level category from the object in the display; in study 2, the invalid labels denoted an unrelated object from the **same** superordinate-level category as the object in the display. The procedure was the same in the control study except that no word labels were presented before the bipartite test displays. (See Fig 4A)

## Drift diffusion model

**The basic drift diffusion model.** The basic drift diffusion model (DDM) models decision making between two options as an evidence accumulation process (Fig 3A). In this view, the model integrates noisy information about the decision over time to form a decision variable $y$ that captures the relative evidence for option 1 (e.g., right) vs option 2 (e.g., left); $y > 0$ denotes that the model has more evidence in favor of option 1, $y < 0$ denotes that the model has more evidence in favor of option 2. During the decision process, the evidence starts at some initial value, $x_0$, that captures the initial bias in favor of one option of the other, if any, and evolves over time according to the following stochastic differential equation:

$$dy = vdt + dn \tag{1}$$

where $dy$ denotes the instantaneous change in evidence, $dt$ denotes a small unit of time, $v$ denotes the drift rate (the average signal in favor of one option or the other) and $dn$ denotes Gaussian random noise of mean 0 and variance $s^2 dt$. Simulating Equation 1 leads to the evidence taking a drifting random walk over time, where the variance of the walk is

governed by the noise parameter $c$ and the drift by the drift rate $v$. It is therefore typical in the DDM literature to set the noise parameter $s = 1$ and interpret the drift rate $v$ as a signal-to-noise ratio (i.e., setting the noise parameter $s$ to 1, the ratio $v/s$ simplifies to $v$). With $s$ standardized to 1 across different studies, the drift rate $v$ can now be directly interpreted as the signal-to-noise ratio, which makes the model more interpretable and standardizes comparisons across studies, this is a normalization step that simplifies the model without changing its fundamental characteristics).

The DDM makes a decision when it crosses one of two thresholds, choosing option 1 if it crosses the top threshold at $+a$ and option 2 if it crosses the bottom threshold at $-a$. The time at which it crosses the threshold corresponds to the decision time (DT), which is related to the response time (RT) via

$$RT = DT + T_{er} \tag{2}$$

where $T_{er}$ is a parameter of the model corresponding to the amount of time not related to the decision – e.g., sensory processing time involved in encoding the stimulus (e), motor preparation time involved in responding to the stimulus (r).

Overall, the DDM has five free parameters: the initial condition, $z$; the drift rate, $v$; the noise, $s$, the threshold, $a$; and non-decision time $T_{er}$. Ideally, we would fit all five parameters to the behavioral data. Unfortunately, however, because some of these parameters only appear as ratios in expressions for accuracy and response time, only four parameters can be independently estimated from data. Because the parameters of the drift diffusion model are only identifiable up to a scale factor [35], the noise parameter ($s$) is typically set to 1. To apply this model to our tasks, we next had to make assumptions about how these parameters might change in different task conditions. Note that we use Ratcliff's notation [45] in the current paper, but we want to acknowledge to readers that we used Bogacz's notation [35] in our original code. The correspondence between Ratcliff's notation and Bogacz's notation is detailed in S1 Text.

**Application of the drift diffusion model to the control experiment.** First, we applied the DDM to the simpler control experiment. In this experiment, familiar objects can appear on the left or right side of the border (side, $S = +1$ for right, -1 for left) and in an upright or inverted orientation (orientation, $O = +1$ for upright, 0 for inverted). We assumed that side and orientation could affect the drift rate in the model, specifying an equation for the drift rate as

$$v = v_{base} + v_s S + v_{so} SO \tag{3}$$

Where $v_{base}$ describes the baseline drift rate corresponding to a drift bias to left or right, $v_s$ denotes the main effect of side and $v_{so}$ denotes how the effect of side is modulated by orientation.

For the rest of the parameters, the threshold $a$, the starting point $z$, and non-decision time $T_{er}$, were assumed to be constant regardless of the side and orientation of the stimulus. This assumption, that the threshold, starting point, and non-decision time are independent of the stimulus, is common in drift diffusion models of behavior and reflects the idea that threshold and starting point are set before the stimulus is presented [35,37]. Thus, for the control task, our model has 6 free parameters that we fit to the responses and response times of the participants: $v_{base}$, $v_s$, $v_{so}$, $z$, $a$, and $T_{er}$.

**Application of the drift diffusion model to the main task.** To extend our model to capture the effects of word labels, we also included the effect of the word label *type* ($L = +1$ for valid, 0 for invalid) on the drift rate, starting point, and threshold. Thus, for the main task, the equation for drift rate becomes:

$$v = v_{base} + v_s S + v_{so} SO + v_{sp} SL + v_{sol} SOL \tag{4}$$

Where $v_{sp}$ denotes the change in drift rate when a valid word label is present and $v_{sol}$ denotes the change in drift rate when a valid word label is present and the stimulus is upright (As mentioned before, $L = 0$ for invalid).

Additionally, to determine whether valid labels lead to faster response times independent of evidence integration, we allowed for the possibility that labels can affect the non-decision time.

$$T_{er} = T_0 + T_l L \tag{5}$$

Where $T_0$ denotes the baseline non-decision time and $T_l$ is the change in non-decision time for a valid label (Again, $L=0$ for invalid).

Finally, for starting point and threshold, we still assume they are constant regardless of labels. Thus, our model for the task has 9 free parameters, $v_{base}$, $v_s$, $v_{so}$, $v_{sl}$, $v_{sol}$, $z$, $a$, $T_l$ and $T_0$.

## Model fitting

We fit the DDM to behavior from Skocypec and Peterson's studies [19] using a maximum likelihood approach. In particular, we find the parameter values that maximize the log likelihood of observing the choices and response times on all trials $i = 1, \ldots, N$.

$$LL(\theta) = \sum_{i=1}^{N} \ln L_i(\theta) \tag{6}$$

where the individual trial likelihood is of observing choice $C_i$ made after response time $T_i$ is:

$$L_i(\theta) = f_{ddm}(C_i, T_i | \theta) \tag{7}$$

Where $\theta$ denotes the parameters of the model and $f_{ddm}$ is the first passage time density computed using the method proposed by Navarro and Fuss [46]. The MLE procedure was run in MATLAB by minimizing the negative log-likelihood function using the *fmincon* function. Data and code to replicate these analyses are available at https://osf.io/g3mxb/.

With so many free parameters in the model an obvious concern is that they will not be identifiable from data. To test whether this was the case, we performed a parameter recovery analysis [37]. The results, shown in Figs A and B in S2 Text, showed an excellent correspondence between the fitted and ground truth value for each parameter suggesting that all parameters are identifiable in our task. Therefore, we further conducted parameter comparisons across different conditions to investigate the effect of semantic labels.

## Results

### Basic behavior

As can be seen in Fig 5, after fitting, the drift diffusion model captures all the main effects in both the control and experimental studies. (All the reaction times shown in Fig 5 combine correct and error responses.) In the control labels-absent studies, the model captures the more accurate responses in the upright condition (i.e., the orientation effect) without any orientation-dependent RT differences. In both labels-present studies, the model captures the orientation effect and faster and more accurate responses following valid labels. Thus, at least qualitatively, the model captures all the main effects in the data. We next examine the evidence regarding condition-dependent drift rate, evidence threshold, and starting point to reveal the mechanisms of Skocypec and Peterson's results [19].

### Drift rates

To determine whether valid labels increase drift rate, invalid labels reduce drift rate, or both effects occur, drift rates obtained in the labels-present studies (studies 1 and 2) were first compared to drift rates obtained in the control labels-absent study. Control drift rates were 0.975 and 0.804 for upright and inverted displays, respectively, a statistically significant difference, $p=0.001$. For each of the studies, a 2 (study: 1 [or 2] vs. control) x 2 (orientation: upright vs. inverted)

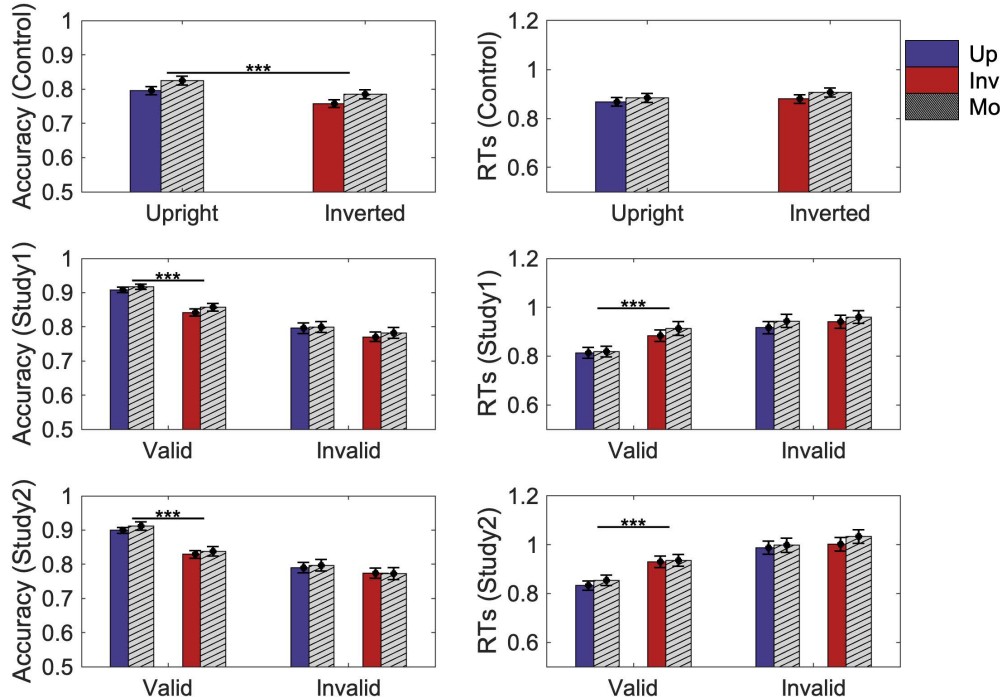

**Fig 5. Accuracy (column 1) and response times (RTs) in seconds (column 2) for both experimental data and data simulated from the model for both control (row 1) and experimental studies 1 and 2 (rows 2 & 3, respectively).** Blue bars represent the upright condition("Up"). Red bars represent the inverted condition ("Inv"). Filled bars represent Skocypec & Peterson's [19] behavioral data; striped bars represent data generated from the analytical expression of the model "Mo"). Error bars represent pooled standard errors. *** indicates $p < 0.001$.

mixed ANOVA was conducted for drift rates following valid and invalid labels separately. To preview: our drift diffusion model results show that valid labels increase drift rate over control whereas invalid labels did not affect drift rates. Fig 6 shows the drift rates obtained in the labels present studies; Table 1 shows the increment over control.

**Valid labels versus no labels.** In both study 1 and study 2, drift rates were significantly higher when valid labels preceded the displays compared to when no labels preceded the displays in the control study (See Table 1) and the increase was larger for the upright than inverted displays (see Fig 6, top row). For both studies these effects were evident in main effects of orientation and study and an interaction between orientation and study. Drift rates were higher in the upright condition than the inverted condition (study 1: $F(1, 216) = 104.72$, $p < 0.001$, η2 = 0.33; study 2: $F(1, 214) = 105.95$, $p < 0.001$, η2 = 0.33). Drift rates were higher in the labels-present studies than in the control study (study 1: $F(1, 216) = 50.78$, $p < 0.001$, η2 = 0.19; study 2: $F(1, 214) = 28.62$, $p < 0.001$, η2 = 0.12). For both studies 1 and 2, the interaction between study and orientation reached significance (study 1: $F(1, 216) = 28.65$, $p < 0.001$, η2 = 0.12; study 2: $F(1, 214) = 29.53$, $p < 0.001$, η2 = 0.12). Simple effects analyses indicated that the increment in drift rate following a valid label was larger in the upright than in the inverted condition, for study1, $t(104) = 6.07$, $p < 0.001$; for study 2, $t(102) = 6.12$, $p < 0.001$. The evidence that following valid labels the increment in drift rate over the control condition is greater for upright than inverted displays supports the Semantic Network hypothesis rather than the Label-feedback hypothesis where orientation effects are not predicted.

**Comparison of drift rates following valid labels in study 1 and study 2.** To examine whether drift rates on valid label trials were affected by the context shaped by the invalid labels shown on other trials in the labels-present studies, we conducted a 2 (study: 1 vs. 2) x 2 (orientation: upright vs. inverted) mixed ANOVA on valid drift rates. Drift rates following

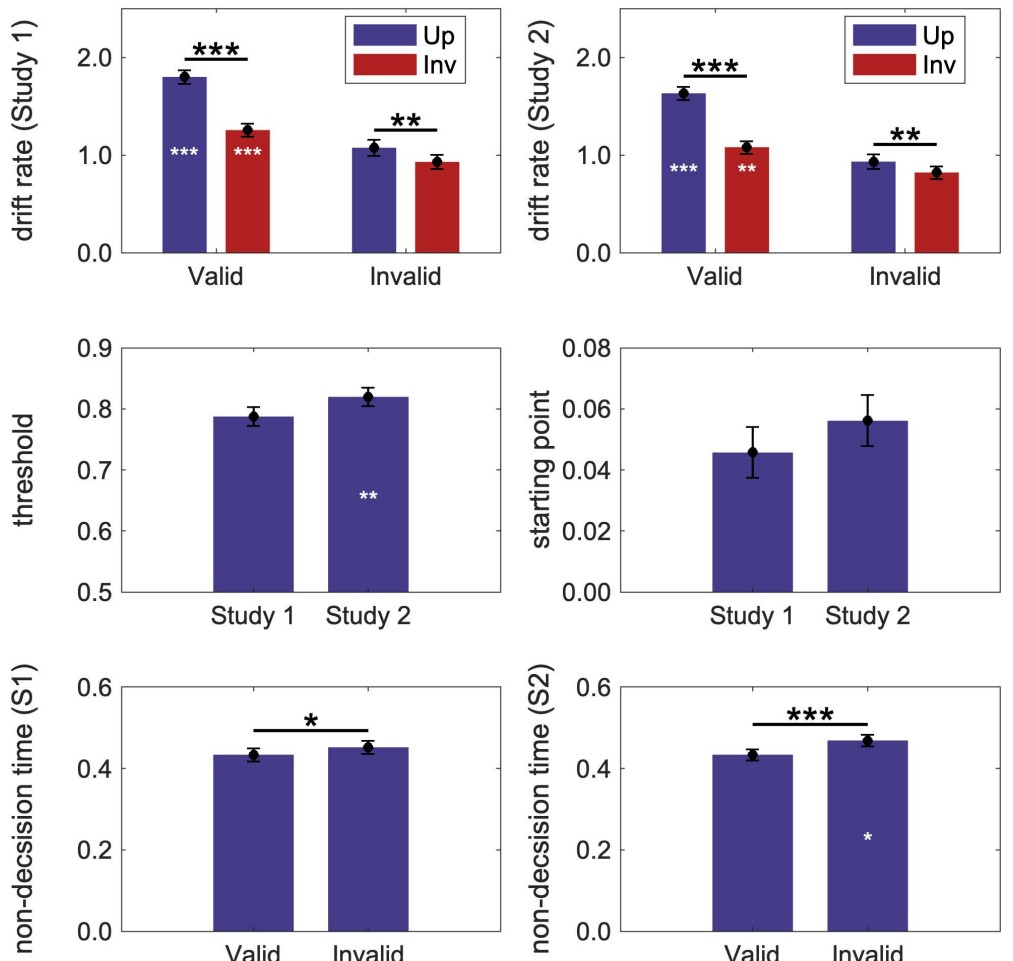

**Fig 6. Model parameters in labels-present studies.** (First row) drift rate (v). (Second row left) threshold (a). (Second row right) starting point (z). Note the different scales on the ordinates of the figures in the second row. (Third row) non-decision time ($T_{er}$). Higher values indicate a greater drift rate/threshold/starting point/non-decision time. For starting point, positive values indicate starting points toward the right side, negative values indicate starting points toward the left side. (Left column) study 1; (Right Column) study 2. White asterisks indicate statistically significant increments over control. Error bars represent pooled standard errors. *** = $p < 0.001$, ** = $p < 0.01$, and * = $p < 0.05$.

**Table 1. Increments of drift rates in studies 1 and 2 over control.**

| Study | Orientation | Increment | 95% CI | p-value |
|---|---|---|---|---|
| Study 1 | Upright | .825 (.101) | [0.626, 1.024] | < 0.001 |
| | Inverted | .451(.091) | [0.272, 0.630] | < 0.001 |
| Study 2 | Upright | .657 (.099) | [0.461, 0.853] | < 0.001 |
| | Inverted | .275 (.088) | [0.101, 0.449] | 0.002 |

Note: Mean increment over control by orientation condition in studies 1 and 2 (standard deviations in parentheses), confidence intervals (CI), and *p*-values for the comparisons to control.

valid labels were higher in study 1 ($M = 1.527$) than in study 2 ($M = 1.356$), $F(1, 206) = 4.03$, $p = 0.046$, $\eta2 = 0.02$ (see Fig 6, first row). This small, but statistically significant effect indicates that the rate of evidence accumulation following valid labels is affected by the context established by the type of invalid labels present on other trials within a study. In study 2, the semantic network of the target object overlaps with the semantic network of the object denoted by invalid labels because they are both in the same superordinate-level category. This overlap, present on 50% of the trials, may introduce enough noise in the evidence accumulation process to cause the reduction in drift rate observed on valid trials.

**Invalid labels versus no Labels.** Mixed ANOVAs comparing drift rates following invalid labels to those observed in the control labels-absent study were conducted separately for study 1 and 2. The factors were study ([1 or 2] vs. control) and orientation (upright vs. inverted). For both studies, this comparison revealed main effects of orientation: drift rates were higher when displays were upright rather than inverted (study 1: $F(1, 216) = 28.49$, $p < 0.001$, $\eta2 = 0.12$; study 2: $F(1, 214) = 21.79$, $p < 0.001$, $\eta2 = 0.09$). No interaction between study and orientation effect was observed in either study 1, $F(1, 216) = 0.20$, $p = 0.660$, or study 2, $F(1, 214) = 0.96$, $p = 0.329$. We also compared drift rates following invalid labels between study 1 and study 2 with a Mixed ANOVA, the results showed a main effect of orientation, $F(1, 216) = 13.73$, $p < 0.001$, $\eta2 = 0.06$. No other significant effect was observed, $ps > 0.205$.

## Threshold

Because the orientation of the objects in a test display and the validity of the label preceding it was unknown until after the test display appeared, we assumed that threshold did not vary with condition within an experiment, although it could vary across experiments. (Participants were not required to recognize the objects, only to detect them. Indeed, in the brief masked exposures they may not have recognized the objects or noted the correspondence between upright and inverted stimuli.) The mean evidence thresholds were 0.754, 0.787 and 0.820 for the control study, study 1 and study 2, respectively.

For study 1, the threshold did not differ from control, $t(216) = 1.67$, $p = 0.097$. In contrast, the threshold for study 2 was significantly higher than control, $t(214) = 3.42$, $p = 0.001$, *Cohen's d* = 0.46 (Fig 6, left middle panel.) Thus, more evidence was required before the detection response in study 2, when the invalid labels denoted unrelated objects in the same superordinate category as the target object, than in the control study; this was not the case in study 1 when the invalid labels denoted unrelated objects in a different superordinate category from the target object. We attribute the higher threshold in study 2 relative to control to noise in the semantic network when the objects denoted by the labels and target objects overlap. When this happens, more evidence must be accumulated from the display before detection occurs. This is not the case when the semantic networks activated by the labels and target objects do not overlap (i.e., when invalid labels denote objects in a different superordinate category). That detection threshold is affected by noise in the semantic system is consistent with Skocypec and Peterson's [19] proposal that the detection of well-known objects like those depicted in the bipartite displays is not just *affected by* semantic activation; it *entails* semantic activation.

## Starting point

The starting points in all studies revealed a small, but statistically significant, bias to detect objects on the right side of the central border, $ps < 0.001$. The right-side bias did not vary with study, however (Fig 6, right third row). For study 1 compared to control, $p = .507$; for study 2 compared to control, $p = 0.970$; for study 1 compared to study 2, $p = 0.539$. Given that objects were depicted equally often on the left and right sides of the displays, participants slight, condition independent, preference to detect objects on the right side may reflect a bias to press the right button when their attention lapsed.

## Non-decision time

As can be seen in the bottom row of Fig 6, non-decision time was longer on invalid than valid trials in both Study 1, $t(104) = -2.18$, $p = 0.031$, and Study 2, $t(102) = -4.38$, $p < 0.001$. Next, we conducted independent-sample t-tests to compare

non-decision times in both the valid and invalid conditions of Studies 1 and 2 to non-decision times in the control group. To control for multiple comparisons, we applied a Bonferroni correction, adjusting the significance level to $p = 0.025$. $T_{er}$ for Study 1 was not different from control: $t(216) = 1.36$, $p = 0.177$ for the valid labels condition and $t(216) = 0.42$, $p = 0.675$ for the invalid labels condition. $T_{er}$ for Study 2 was longer than for the control group for the invalid labels condition, $t(214) = 2.35$, $p = 0.020$, *Cohen's d* $= 0.32$, but not for the valid labels condition, $t(214) = 0.47$, $p = 0.642$. Because $T_{er}$ was longer than control only on invalid trials, we attribute the difference between $T_{er}$ on valid and invalid trials to processes operative on invalid trials. One possible reason for the longer $T_{er}$ following invalid labels in study 2 is that at the time of response, participants were less confident in the invalid same-superordinate category label condition of study 2 than in the control no-labels study. Other research [47] has linked longer non-decision time with lower confidence.

Non-decision time differences should be evident in the fastest responses. Therefore, to further examine $T_{er}$, we compared the fastest 20% of RTs between valid and invalid conditions in study 1 and 2 using a 2 (study: 1 vs. 2) x 2 (label: valid vs. invalid) mixed ANOVA. The results showed a main effect of label type: the fastest RTs were longer in the invalid condition than the valid condition, $F(1, 206) = 99.42$, $p < 0.001$, $\eta 2 = 0.33$ (see S1 Fig); and an interaction between label type and study, $F(1, 206) = 4.58$, $p < 0.001$, $\eta 2 = 0.02$: the difference between the fastest valid and invalid RTs is larger in study 2 (-0.068 s) than in study 1 (-0.044 s). Hence, the analysis of fastest RTs supports the $T_{er}$ results returned by the DDM.

## Discussion

Our goal in the present study was to elucidate the mechanisms underlying the results of Skocypec and Peterson's investigation [19] of whether valid and invalid labels shown before test displays affect object detection per se and not feature detection. To do so, we applied a drift diffusion model (DDM) to individuals' response times on all trials regardless of whether their responses were accurate or inaccurate. Recall that Skocypec and Peterson [19] had analyzed individuals' mean responses on accurate trials only. The DDM derived three potential mechanisms for how word labels affect object detection: (1) By changing the drift rate, i.e., the rate at which evidence for the detection response accumulates (i.e., processing speed); (2) By changing the evidence threshold, i.e., the detection boundary representing the amount of evidence that must be accumulated from the display before a detection response is made; (3) By changing the starting point for a decision, i.e., whether one of two potential responses is favored at the outset; (4) By changing the non-decision time. Analyzing the full set of results, we found that, compared to the control studies without labels, labels influenced object detection through altering drift rate, threshold, and non-decision time. No changes in starting point were observed.

### Drift rate

Compared to a control labels-absent study, valid labels shown before test displays increased the rate of evidence accumulation for the object in the display; the increase was larger for upright than for inverted objects. Invalid labels had very little influence on behavior. This is exactly the pattern predicted if words activate a semantic network that includes the neural population representing the denoted object. We propose that the prior activation of the test object's neural population reduces the number of reentrant cycles needed to localize the object to the right or left of the central border (see discussion of Fig 7 below).

Valid labels that denoted the object in the test display at a basic level (i.e., bell, woman, etc.) were present on half of the trials in the labels-present studies. On the other half of the trials, invalid labels denoted unrelated objects. In study 1, invalid labels denoted objects in a **different** superordinate-level category than the object in the test display. In study 2, invalid labels denoted objects in the **same** superordinate-level category as the object in the test display. The drift rate following valid labels was affected by the context provided by the type of invalid labels shown on the other half of the trials: it was lower in study 2 in the context of invalid labels denoting an object in the same, rather than a different, superordinate-level category as the test object as in study 1. We propose that activating another object in the same

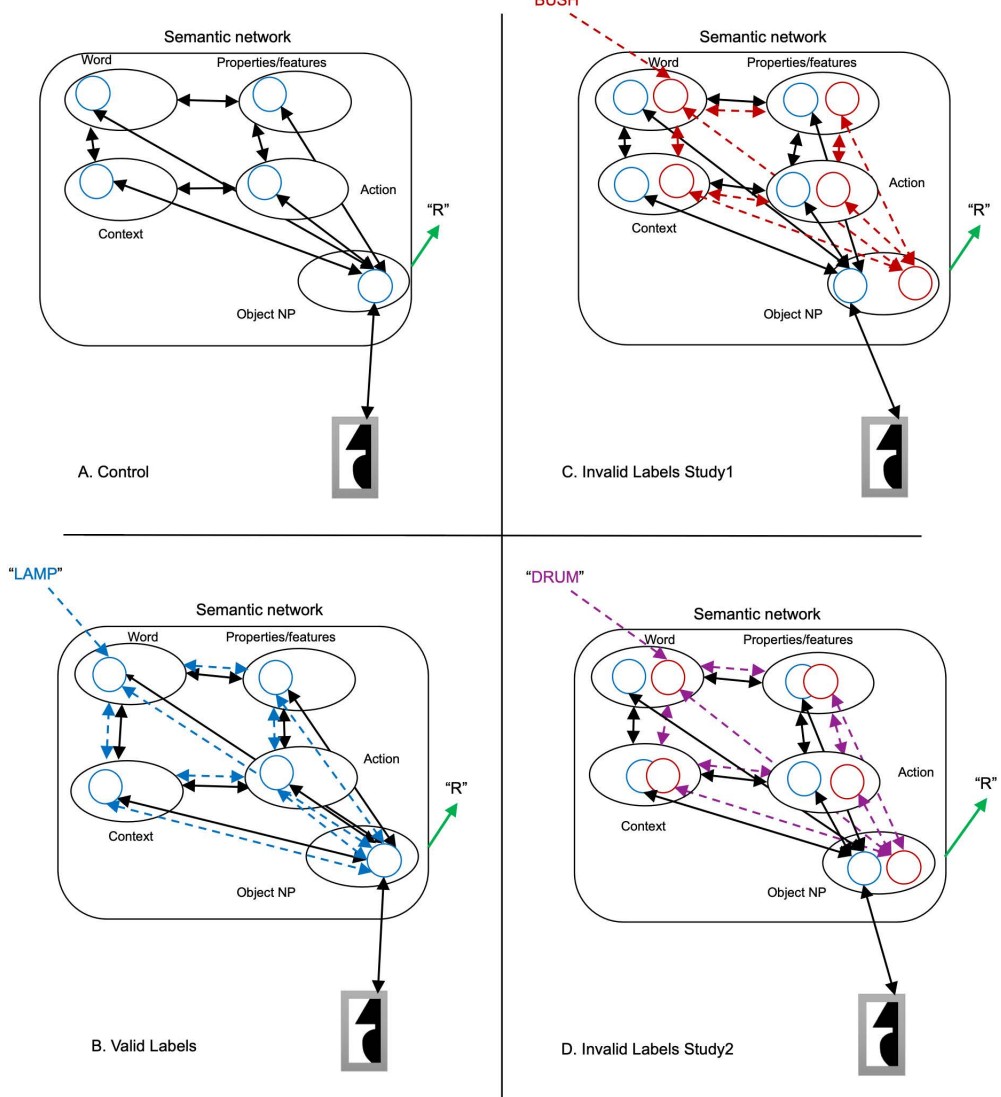

**Fig 7. Potential mechanisms operative in Skocypec and Peterson's object detection task [ 19].** Solid black lines with double-headed arrow endings indicate reentrant activity initiated by the object in the bipartite test display, both within the semantic network and between the semantic network and a lower-level representation of the test display (shown below the semantic network). Dashed lines with double-headed arrow endings indicate reentrant activation in the semantic network initiated by labels shown before the test displays, blue for valid labels, red or purple for invalid labels. Blue circles in the semantic network indicate semantic representations of the object in the test display in the various portions of the semantic network (e.g., context, object, etc.). Red & purple circles indicate semantic representations of objects denoted by invalid labels. Object NP = neural population representing an object. A) control labels-absent study. B) Valid labels in labels-present studies. (C-D) Invalid labels in labels-present studies; C) different superordinate-level invalid label as in study 1. D) same superordinate-level invalid label as in study 2. The green arrow labeled "R" emerging from the object NP in the semantic network indicates the participant's right or left response; it's shown in black to indicate that although the drift rate and threshold are affected by activity throughout the semantic network which affects activity within the NP representing the object, the R/L response must be dominated by recurrent activity between that NP and a lower-level representation of the object in the display.

superordinate-level category as the target object introduces noise into the decision process because the semantic networks of the two objects overlap. This noise, present on 50% of the trials in study 2, lowered the signal to noise ratio and resulted in an overall reduction in drift rate. These results are the opposite of what would be predicted on the

Label-Feedback hypothesis where larger interference would be expected when labels denote objects with different features (i.e., objects in different superordinate level categories).

**Threshold**

Evidence thresholds were elevated in study 2 relative to the control study but not in study 1; an interpretation in terms of semantic network noise also accounts for these threshold changes: When the semantic network activated by a label shown before the test display and by the object in the test display overlap as they do in study 2, more evidence had to be accumulated from the display to reach the detection threshold than when labels were absent. It takes time to accumulate this additional evidence and detection response times are longer. Thus, threshold indices obtained through modeling support Skocypec and Peterson's claim [19] that object detection entails semantic activation.

Fig 7 illustrates the semantic network implicated by the DDM in label effects on the detection of meaningful objects. The illustration uses a sample test display that depicts a portion of a table lamp on the right side of the central border of a bipartite test display; hence, the accurate response is "Right." Note that object detection responses, indicated by a green arrow and "R", are dominated by reentrant activity between the object representation in the semantic network and a lower-level representation of the object in the display, albeit influenced by activity in the semantic network. Panel A illustrates how object detection decisions are made in the control labels-absent study: The critical region in the bipartite display initiates activity in the neural population (NP) representing the object in the display (here, a lamp); see the blue circle in the Object NP portion of the semantic network. The NP includes more representations of the object in its typical, or upright, orientation than in an inverted, upside down, orientation and therefore accumulates evidence faster for upright than inverted objects [25]. As evidence accumulates in the Object NP, activity spreads to other components of the semantic network associated with that object (e.g., words denoting that object, object properties and features, contexts in which that object is likely to occur, actions related to the object, etc.); see the blue circles in other parts of the semantic network. Reentrant activity within the semantic network enhances activity throughout the network (shown by double-headed black arrows in the semantic network). Reentrant processes from the Object NP to lower retinotopic areas are necessary to localize the object on one side of the border (see double-headed black arrow between the lower-level representation of the critical region in the display and the object portion of the semantic network). A right/left side response is generated when the activity in the object NP exceeds threshold for one side of the display.

Panel B depicts reentrant activity in the semantic network following presentation of valid labels (e.g., the label "lamp" for the object in the test display. The label pre-activates the semantic network associated with the object it denotes including the Object NP (dashed blue double headed arrows). Double-headed black arrows indicate activation initiated by the object in the test display, as in A. Converging input to the Object NP from the semantic network and the input reduces the number of reentrant cycles necessary to generate the R/L response required by the task, thereby accounting for higher drift rates than control following valid labels.

In Panel C, dashed red lines illustrate activity in the semantic network following an invalid label denoting an object in a **different** superordinate-level category from the object in the test display (e.g., the label "bush" shown before the test display depicting a lamp). Semantic representations of bush throughout the semantic network are shown in red circles. Activation spreads throughout the semantic network, including to the Object NP for bush. The different colors of and spacing between the circles for lamp and bush illustrate that there is little overlap between the representations of objects in different superordinate-level categories. Accordingly, pre-activation of the semantic network by a label denoting an unrelated object in a different superordinate level category doesn't affect the rate of evidence accumulation for the object in the test display.

In Panel D, dashed purple lines illustrate activity in the semantic network following an invalid label denoting an object in the **same** superordinate-level category as the object in the test display (e.g., the label "drum" shown before the test

display depicting a lamp). Semantic representations of drum throughout the semantic network are shown in purple circles. Activation spreads throughout the semantic network, including to the Object NP for drum. The similar colors of and spacing between the circles for drum and lamp illustrate overlap in the semantic networks of objects in the same superordinate-level category which produces noise. It is this noise that reduces drift rate and raises thresholds when same-superordinate level category invalid labels are used in study 2.

### Object detection entails semantic activation: Conflict or noise?

Skocypec and Peterson [19] proposed that in study 2 a conflict existed between the similar semantic networks activated by the invalid label and by the object in the test display. Our DDM model suggests a way to think about this conflict: There is more noise in the system in study 2 where invalid labels denote another object in the same superordinate-level category as the object in the test display. Greater noise lowers the signal-to-noise ratio (i.e., decreases drift rate). Indeed, we found that drift rate is lower following valid labels in study 2 than in study 1. Greater noise also increases threshold (i.e., more evidence must be accumulated from the display to differentiate the semantic networks activated by the label and the object). Indeed, we found threshold is higher in study 2 than in the control study; no such difference was found in study 1. On this "noise" account, a separate conflict resolution process is not necessary. An interesting question is whether threshold adjustments are a purely bottom-up response to increased noise or reflect top-down control. Future research can adopt more complicated models to investigate this question.

The larger $T_{er}$ on invalid trials in study 2 compared to control may reflect uncertainty before decision due to the noisy semantic activation. We attribute $T_{er}$ to a factor affecting response execution rather than stimulus encoding both because $T_{er}$ is longer than control only on invalid trials in study 2 and because an effect due to stimulus encoding time should have been larger in study 1 where the features of the objects denoted by the invalid label differed from those of the object in the test display.

### Sensitivity of the DDM

In addition to revealing the mechanisms of Skocypec and Peterson's results [19], DDM modeling revealed some results that were not evident in their separate analyses of participant's mean accuracy and response times per condition, specifically that

(1) drift rate is higher in the control condition for upright than inverted displays. Skocypec and Peterson [19] were perplexed by observing that their control trials showed an orientation effect in accuracy but not in response time because it seemed that both effects were predicted by an explanation in terms of neural populations. Using RTs obtained regardless of accuracy to assess the time needed to accumulate sufficient evidence for a response (drift rate) removes any questions and reveals drift rate to be an excellent index of visual processing.

(2) drift rate following valid labels was affected by the context of the type of invalid labels that were present on half of the trials; this context effect is consistent with prevalent sequential trial effects [48,49]. It would be interesting to explore the span of trials over which the effects of invalid labels persist.

### Label-feedback hypothesis versus the semantic network hypothesis

Can the Label-Feedback hypothesis be altered to accommodate our results by supposing, for instance, that feature representations are orientation dependent? The classic view in perception research, supported by much research, is that visual features are orientation independent [20,50], whereas orientation dependence is a hallmark of configuration effects. What about supposing that the Label Feedback hypothesis extends to features in context? We don't see how this would help, as features in context are familiar configurations, which are not low-level representations. In the Label-Feedback hypothesis,

feedback from high-level word representations extends to low-level features of objects [18]. Moreover, in other experiments, we have found that that the same features arranged in novel configurations do not operate as object priors, even when sub-configurations comprising familiar parts are intact [51]. Finally, our evidence contrary to the Label-Feedback hypothesis extends beyond orientation dependency: Drift rates were lower when invalid labels denoted an object in the same, rather than a different, superordinate-level category as the object in the display; and thresholds were higher than control only when the invalid label denoted an object from the same superordinate-level category as the displayed object, not when it denoted an object from a different category. Indeed, in study 1, invalid labels denoting objects in a different superordinate-level category had no effect on either drift rate or thresholds. On the Label-Feedback Hypothesis, the opposite pattern should have been observed because the features of objects in a different superordinate-level category differ more from the features of the objects in the display. Under this hypothesis, a larger discrepancy would require more prediction revision; updating predictions would take time and would prolong internal processing, therefore lowering drift rates or elevating thresholds. The absence of these predicted effects suggests that the Label-Feedback Hypothesis cannot easily be modified to account for our results. The pattern of results is better fit by the Semantic Network Theory of Label effects. We note, however, that an independent t-test comparing thresholds between study 1 and study 2 showed no difference, $p = 0.140$. Future research is needed to investigate whether a small amount of semantic interference occurs even in Study 1.

## Limitations of this research

One limitation of the current research is that our model (DDM) tends to overestimate RTs and ACC in the last quantile (albeit slightly) in all studies (control, S1 and S2). As shown in the quantile-quantile plot in the Fig A in S3 Text, this discrepancy was mainly caused by the last quantile of correct trials. For this discrepancy, we consider 2 possibilities, 1) Model Limitations: The standard DDM assumes consistent decision-making processes across all trials. However, in the slowest correct trials, participants may engage in additional cognitive processes not captured by the model, such as increased deliberation or occasional lapses in attention that are subsequently corrected before responding. 2) Speed-Accuracy Trade-Offs: Participants might collapse their decision thresholds dynamically. To understand how good the fit to the data by the model was, we first tested the RT deviation by comparing the simulated RT to the actual RT using independent t-tests for each quantile. We found no significant differences between simulation and actual RT, $ps > 0.07$. Next, as shown in Fig B in S3 Text, we plotted the accuracy against RT quantile for both simulation and actual data for all conditions in the three studies. For each condition, we used independent t tests to examine the difference between simulated and actual data. While some of the conditions showed a difference only in the 4th quantile (see black asterisks in Fig B in S3 Text), most of the simulations are aligned with the actual data and none of the differences evident in studies 1 and 2 was larger than those evident in the control study, $ps > .24$. Therefore, we suggest that the model captures the overall decision-making process, especially the first 3 quantiles of the data. However, it is possible that threshold adjustments could happen in some of the conditions in the fourth quantile, which could indicate a strategic adaptation to the task environment. On this perspective, the increased threshold in study 2 would reflect a strategic top-down control rather than a purely passive response to increased noise. Future research can use collapse bound DDM or by fitting to model to successive blocks to investigate this point.

Another limitation of our research is the model's inability to explicitly distinguish whether the decision-making process involves one or multiple cognitive processes or stages. Traditional sequential sampling models like DDMs often assume a single integrated process of evidence accumulation [35,52]. However, this may oversimplify the neural processes involved, especially when considering reentrant neural processing loops in perception. In the current experiment, the perception of an object and its spatial location (R/L of a border) are not considered strictly linear or sequential processes but are presumed to involve iterative feedback loops. More specifically, detecting the object's identity might be influenced by how its position is interpreted, and vice versa, with the brain constantly refining its interpretation as more information was

processed and integrated in neurons in the LIP reflecting decision [44,53]. To resolve this limitation, future studies could incorporate more complex models that allow for multiple stages of decision-making, such as the hierarchical drift diffusion model or models with interactive competitive dynamics. These approaches can accommodate the parallel processing and feedback mechanisms inherent in neural computation.

Bornstein et al [54] used a multi-stage DDM (MSDDM) to model their results. They manipulated the validity of pre-cues (50%, 60%, 70%, 80%) and the quality of the target stimuli, which were faces or scenes. Their results were best explained by a MSDDM, but their procedure was quite different from ours. The closest condition was a 50% valid cue condition followed by a high-resolution test display. Unfortunately, they did not analyze their results from this condition, so their article is not informative regarding what might be learned from a MSDDM model of our results.

## Supporting information

**S1 Text. Correspondence between Bogacz's and Ratcliff's notation.**
(DOCX)

**S2 Text. Parameter recovery.**
(DOCX)

**S3 Text. Quantile analysis.**
(DOCX)

**S1 Fig. Comparison of reaction times between fastest 20% valid and invalid trials.** *** indicates $p < 0.001$. Red asterisks represent interaction between label type and study.
(JPG)

## Author contributions

**Conceptualization:** Mary A. Peterson.

**Data curation:** Jingming Xue.

**Formal analysis:** Jingming Xue.

**Methodology:** Jingming Xue, Robert C. Wilson.

**Supervision:** Robert C. Wilson, Mary A. Peterson.

**Visualization:** Jingming Xue.

**Writing – original draft:** Jingming Xue, Robert C. Wilson, Mary A. Peterson.

**Writing – review & editing:** Jingming Xue, Robert C. Wilson, Mary A. Peterson.

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
