## [Decision Letter · Decision Letter 0]

Dear N/A Xue,

Thank you very much for submitting your manuscript "Semantic Influences on Object Detection: Drift Diffusion Modeling Provides Insights Regarding Mechanism" for consideration at PLOS Computational Biology.

As with all papers reviewed by the journal, your manuscript was reviewed by members of the editorial board and by several independent reviewers. As you will see, the reviewers found the work interesting, but raised several significant issues that make the paper, as written, difficult to evaluate. In particular, there was concern that the modelling in its current form does not clearly contribute to our understanding of the phenomenon, and would require significant clarification and likely significant reevaluation. If you feel you can address the reviewer comments, I am willing to consider a revised version of the manuscript. The paper would then need to be sent back to the reviewers (and possibly new reviewers), who would need to reassess the work. I cannot, of course, guarantee that the resubmitted manuscript would be successful. 

Sincerely,

Alex Leonidas Doumas

Academic Editor

PLOS Computational Biology

Andrea E. Martin

Section Editor

PLOS Computational Biology

Reviewer's Responses to Questions

**Comments to the Authors:**

Reviewer #1: The authors present a reanalysis of data reported previously by Skocypec and Peterson (2022) using Ratcliff’s diffusion decision model (DDM). In the original study, participants viewed bipartite stimuli showing familiar objects. Images were presented either in a familiar upright orientation or were inverted. For some participants, object presentation was preceded by a basic-level category label that either matched the stimulus (valid label) or mismatched (invalid label). Valid and invalid labels were randomly intermixed and comprised half of the label-present trials each. Across studies, the invalid labels could be drawn from different superordinate categories (Study 1) or the same superordinate category (Study 2). A control study was used as a baseline condition, where no labels were presented at all (Study 3).

The authors found that the DDM was able to successfully reproduce the key qualitative features of the data. Investigation of how parameters varied across conditions revealed both drift rate and threshold effects. Upright images were associated with higher drift rates than inverted images. Valid labels were associated with higher drift rates and invalid labels. Decision thresholds were found to be higher for invalid labels when the labels were drawn from different superordinate categories (no difference was found for same superordinate categories).

I think the authors’ use of the DDM here is innovative and that the study addresses an interesting theoretical question using a more sensitive method of analysis over Skocypec and Peterson’s original study. I do, however, have some major concerns about how clearly the two competing hypotheses can be distinguished—especially in relation to the inversion effect—but also in terms of model parameters. My more immediate concerns are to do with the way the model was parameterized. As detailed below, I do not think it is appropriate to allow threshold (and start-point) to vary as a function of label condition. If my concerns are valid, then this also means that the inferences about the other model parameters (particularly drift rate) are called into question due to potentials for trading off in fitting the data. This makes it extremely challenging to evaluate the potential theoretical contribution of the work at this stage.

Taken together, these concerns are enough to recommend against publication of the work in its current form. At minimum, I think the modelling analysis would have to be redone from the ground up (i.e., disallowing thresholds to vary with label). I do, however, think that the authors might also benefit from allowing non-decision to vary across conditions—a parameter that they constrained to be equal across conditions—as this provides a different avenue for potentially capturing differences between the label-feedback and semantic network hypotheses.

Below I detail my concerns roughly in the order in which they arose during my reading of the manuscript.

1 – I appreciate the care taken to differentiate the label-feedback and semantic network hypotheses, but I was not entirely convinced that the predictions regarding the inversion effect were as clear-cut as the authors made them out to be. On the one hand, I would expect a strong inversion effect for objects encountered in unfamiliar orientations, per the semantic network hypothesis. On the other hand, I would also expect similar disruption under the label-feedback hypothesis since the low-level features activated would presumably be in proportion to the typicality of their presentation. I would expect a feature presented in an unusual context to be more weakly associated with the object representation than I would the same feature presented in a common context. Put another way, are the authors assuming “features” are represented in an entire context-independent manner? I worry this assumption is too strong because it divorces feature representations from potentially useful configural information that would presumably exist at some level of the representational hierarchy.

2 – Diffusion model description – Although I am highly familiar with the use of these models, I found it difficult to follow the model description on pages 9-11 due to the authors’ departure from using conventional notation for the model parameters. For example, drift rate in Ratcliff’s model is usually denoted v (not A), response bias is usually denoted z (not x_0), upper and lower boundaries are usually denoted using +a and -a (or a and 0; not, +Z and -Z). This is a relatively minor issue, but it did affect the fluency with which I could parse the model description. This would not be so bad if the authors’ notation didn’t actively conflict with convention (e.g., as with boundary position and start-point parameters). I think bringing the model description in line with Ratcliff’s standard notation would help improve the readability of this section and make it easier for people to relate the current findings to previous work with the DDM.

Also, I was initially confused by the description of the c parameter (p.9) and the later reference to “an additional noise parameter” (p.11). I later realized on page 15 that there was only one noise parameter, which was c, but prior to that, I assumed there were two noise parameters (one describing across-trial variability in the drift rate and another describing within-trial accumulation noise; the diffusion coefficient). It would be helpful if the authors clarify on page 9 that c is fixed to 1.

3 – Model parameterization – The authors mention on page 11 that all four free DDM parameters are estimated across different label conditions. One concern is that this includes decision threshold (denoted by the authors as z). Traditionally, this parameter is only allowed to vary as a function of task-level factors rather than stimulus-level ones because it is regarded as capturing strategic aspects of how a participant is approaching the task (i.e., it controls their speed-accuracy trade-off).

Because the validity of the label is based on its congruency with the stimulus, allowing threshold to vary with label validity requires threshold setting to be highly reactive—occurring when the stimulus is presented, not when the label appears (cf. the authors’ description of Equation 7 on page 17). To be clear, determining whether a label is valid or invalid in this task context requires some way of evaluating the correspondence between the stimulus and the label, which logically requires identification of the stimulus ahead of being able to locate it on one side of the display.

Minimally, the threshold parameter needs to be fixed across trials. It can, however, be estimated freely across the three studies, which would still provide important information about how participants process information when labels are drawn from the same vs. different superordinate category compared to having no labels at all.

The authors discuss the matter of threshold not varying based on stimulus conditions with respect to left/right side and upright/inverted orientation (pg.17), but label is regarded differently. In this case, however, I think label has to be treated in the same way because its effects can only be realized in relation to the properties of the stimulus.

A similar argument can be made about start-point needing to be fixed across valid/invalid label conditions. Allowing this parameter to vary on a trial-by-trial basis would make sense if the labels were more directly related to the response options (e.g., saying “left” vs. “right”), in which case x0 would make sense as capturing response bias. Here, however, the lack of any direct mapping from label-stimulus congruency to the left/right response makes it questionable as to why/how start-point would be able to vary based on label condition. (That said, I would not expect start-point to vary with label validity since the mapping of validity to left/right is random.)

One point of additional flexibility that the authors did not consider is whether non-decision time might vary based on label condition. Since the major theoretical ideas are based on faster (pre-decisional) activation of an object representation/semantic network, I would think it natural to see if some of the benefits of valid labels might be reflected in faster non-decision times. (A heuristic to see if there might be empirical support for this would be to compare the fastest RTs across valid and invalid conditions—since the fastest RTs determine the positioning of non-decision time in the DDM, this provides a useful clue as to the necessity of freeing up non-decision time.)

4 – The description of the scaling property of the DDM is, I think, slightly inaccurate. The parameters of the DDM are only identified on a ratio scale, which means that one parameter must be chosen to be fixed to an arbitrary value. The authors are correct in pointing out that the diffusion coefficient is usually chosen as the “scaling parameter,” but this is an essential feature of the model, rather than a means of simplifying it. (Apologies if this was the intent of this section and I failed to recognize it! I see that this is explicitly mentioned later on page 16.)

5 – Parameter estimation – I am glad that code have been made available on the OSF, but I think more explicit detail about parameter estimation needs to be provided in text. Was MLE done for individual trials or was it done using a quantile-based approach?

6 – Model fit – The authors provide a visualization of model fit in Figure 5, but the figure is somewhat difficult to read because it requires aligning spatially separated bars. A better visualization would use something like floating symbols overlaid on top of the bars to display model predictions.

Extending on this, it would be nice to see both the data and model predictions for entire RT distributions for correct and error responses using something like a quantile probability plot. I can see that the RT predictions of the model seem to be a little off (e.g., for invalidly labelled inverted stimuli), but presumably these are all showing mean RTs (the figure doesn’t specify) or some other central tendency measure. One question is whether the model predictions are off for (say) the mode of the RT distributions or if the mean is being pulled up by misses in the extreme tails.

Another question is whether the RT data in these panels are combining correct and error responses or if only correct responses are being shown. (Apologies if I have missed this detail.)

7 – Interpretation of Parameter Estimates – This point relates to my concerns about model parameterization (Point #3). I think some of the drift rate effects are quite interesting and potentially important, but I am curious how strongly they were affected by also having freely varying thresholds and starting-points. Since all three of these parameters affect predictions for both accuracy and RT, it is difficult to discern how much stock to place in the drift rate results, for example. The ambiguity introduced by the way the authors parameterized the model has knock-on effects for inferences about individual parameters that make it difficult to evaluate the general theoretical conclusions.

Reviewer #2: This paper uses drift diffusion modelling to investigate the mechanisms underlying a previously published, labelled object detection task. The Label-Feedback hypothesis is that the presence of a valid label pre-activates low level visual features which leads to faster object detection; however, the visual features pre-activated are invariant to orientation. Whereas, the semantic network hypothesis adds that in addition to priming, an invalid label can cause semantic interference depending on the degree of relatedness between the label and object. Also, low level visual features should be sensitive to the frequency of orientation because semantic networks would store that information. In terms of the model, if orientation matters, we expect to see a change in drift rate and, if there is semantic interference, we expect to see noise increase, which should lower drift rates and increase the threshold. The authors find evidence of both effects. The concluding argument, as I understood it, is that semantic interference entails semantic activation; therefore, object detection entails activation.

The paper is well written. In particular, the modelling sections was easy to follow. I think how the computational model demonstrating the interference effect as noise is interesting; however, I think the paper would be stronger if they discussed how else the model might have captured the pattern of behaviour in the original study. Without the clear alternatives, I’m not sure the model could have captured the data any other way; in which case, the contribution of the paper is unclear as it simply reiterates the paper collecting the data.

In my reading, the key contribution of the paper is the demonstration of semantic interference as noise. I am not sure how much the orientation manipulation undermines the Label-Feedback hypothesis, which to my knowledge, is agnostic about the typical frequency of the orientation of the visual feature detectors and, thus, charitably could accommodate this result.

Finally, I’m not sure the concluding argument logically follows. While it’s clear that introducing an expectation via a label allows semantic activation to influence object detection. No evidence is provided that this semantic activation is entailed for object detection without such expectations.

As a technical note, I would have appreciated more details on the parameter recovery analyses, specifically what was the decision rule to suggest that something is recovered and why is that an appropriate decision rule. Some of the correlations are rather small and even for some higher correlations, the scale is off by a wide margin. Why is the scale not important? In light of the importance of noise in the interpretation of the semantic interference results, I think it would be important to illustrate how fixing the scale of the noise changes the other parameters (e.g., as a graph like Figure 3).

**Have the authors made all data and (if applicable) computational code underlying the findings in their manuscript fully available?**

Reviewer #1: Yes

Reviewer #2: None

PLOS authors have the option to publish the peer review history of their article (what does this mean? ). If published, this will include your full peer review and any attached files.

**Do you want your identity to be public for this peer review?** For information about this choice, including consent withdrawal, please see our Privacy Policy .

Reviewer #1: No

Reviewer #2: No
---

## [Decision Letter · Decision Letter 1]

PCOMPBIOL-D-24-01051R1

Semantic Influences on Object Detection: Drift Diffusion Modeling Provides Insights Regarding Mechanism

PLOS Computational Biology

Dear Dr. Xue,

Thank you for submitting your manuscript to PLOS Computational Biology. You will see that the reviewers (and I) were very satisfied with your revision of the manuscript. Reviewer 1 raises some minor points that I think should be addressed (in the paper or if you decide not to, clarified in the paper and addressed in a cover letter detailing why). I do not foresee needing to send this revision back out for review, but, of course, I cannot make any guarantees. 

Please submit your revised manuscript within 30 days Apr 29 2025 11:59PM. If you will need more time than this to complete your revisions, please reply to this message or contact the journal office at ploscompbiol@plos.org. Please include the following items when submitting your revised manuscript:

We look forward to receiving your revised manuscript.

Kind regards,

Alex Leonidas Doumas

Academic Editor

PLOS Computational Biology

Andrea E. Martin

Section Editor

PLOS Computational Biology

**Journal Requirements:**

1) We note that your Manuscript files and Supplementary Figures are duplicated in the online submission form. Please remove any unnecessary files from your revision, and make sure that only those relevant to the current version of the manuscript are included.

2) Please include the affiliation of Dr. Jingming Xue in the online submission form and ensure that it exactly matches with the affiliation listed on the manuscript title page.

**Reviewers' comments:**

Reviewer's Responses to Questions

Reviewer #1: I greatly appreciate the care and thoughtfulness the authors have taken in revising their manuscript. Constraining decision thresholds to vary only as task-level properties rather than stimulus-level properties was necessary to ensure interpretable results. The points of clarification the authors have provided with respect to differentiating the label-feedback and semantic network hypotheses have been helpful and I agree with the authors that the current findings align well with the latter.

I do not have any major concerns with the revised manuscript—the authors have done an outstanding job of addressing my earlier comments. I do, however, have a few relatively minor comments that mostly have to do with the results regarding decision thresholds that I think need further clarification. I address that issue before turning to other more minor points.

1 – Thresholds should be compared across Studies 1 and 2

Currently, threshold parameters for Studies 1 and 2 are compared against thresholds in a no-label control. This is fine to establish that semantic activation is involved in the detection process but it would be more instructive to know how threshold is affected by the degree of activation. Indeed, based on the current results, one would expect to see a threshold difference between Studies 1 and 2 if there is no semantic overlap between targets and invalid labels in Study 1. I note, however, that it would be perfectly understandable if the difference across Studies fails to reach significance—there is likely some low level of activation achieved even by labels for different semantic categories—but it would be instructive to check for.

Another reason for performing a direct comparison of Studies 1 and 2 is that the authors comment on the current results as if a direct study-to-study comparison had been made. For example, on page 33, it is mentioned, “…thresholds were higher when labels denoted an object in the same, rather than a different, superordinate-level category as the object in the display.” This claim can only be supported by comparing thresholds across studies directly.

2 – Are “noise-dependent” changes in threshold strategic or under top-down control?

The argument (e.g., page 32) that additional noise due to semantic overlap of invalid labels and objects increases decision thresholds is an interesting one. The current description of the nature of this effect strikes me as agnostic about the degree to which this is under strategic control (or otherwise influenced by top-down goals). My own view of these kinds of effects is that they likely reflect a strategic adaptation to the task environment, not unlike how changes in response bias can be induced by varying payoffs or the proportion of responses of a certain type required by the task. To the extent that threshold changes adapt to the presence of noise, one could in principle analyze changes in threshold over the course of the task (e.g., by fitting to model to successive blocks/trial epochs). I am not suggesting this can be done for the current study but this could be worth mentioning as a future research direction (e.g., likely requiring a hierarchical approach to model fitting).

3 - Page 13 – “…we can fit the four free parameters of the model…” would be clearer if you said, “…we can estimate the four free parameters of the model…”

4 - Page 22 – The in-text description of the drift rate increments for Studies 1 and 2 is difficult to parse. Consider presenting this information in a Table for better readability.

5 - Page 34 – I think the argument that the data cohere with the semantic network hypothesis better than with the label feedback hypothesis is convincing, but I think the claim about “…more prediction revision … would take longer” needs further clarification as to what is taking longer. I can understand behaviorally that responses will be slower due to (say) increased interference predicted by the label feedback hypothesis, but no explicit cause of what would make things take longer is currently mentioned.

Reviewer #2: I would like to thank the authors for taking the time to address my concerns. I think this is an interesting data analysis.

**Have the authors made all data and (if applicable) computational code underlying the findings in their manuscript fully available?**

Reviewer #1: Yes

Reviewer #2: None

PLOS authors have the option to publish the peer review history of their article (what does this mean? ). If published, this will include your full peer review and any attached files.

**Do you want your identity to be public for this peer review?** For information about this choice, including consent withdrawal, please see our Privacy Policy .

Reviewer #1: No

Reviewer #2: No

**Figure resubmission:**
---

## [Editor Report · Decision Letter 2]

Dear Graduate Student Xue,

We are pleased to inform you that your manuscript 'Semantic Influences on Object Detection: Drift Diffusion Modeling Provides Insights Regarding Mechanism' has been provisionally accepted for publication in PLOS Computational Biology.

Best regards,

Alex Leonidas Doumas

Academic Editor

PLOS Computational Biology

Andrea E. Martin

Section Editor

PLOS Computational Biology

---

## [Editor Report · Acceptance letter]

PCOMPBIOL-D-24-01051R2

Semantic Influences on Object Detection: Drift Diffusion Modeling Provides Insights Regarding Mechanism

Dear Dr Xue,

I am pleased to inform you that your manuscript has been formally accepted for publication in PLOS Computational Biology. Your manuscript is now with our production department and you will be notified of the publication date in due course.

With kind regards,

Anita Estes
